# From magnetic order to valence-change crossover in $EuPd_2(Si_{1-x}Ge_x)_2$ using He-gas pressure

Bernd Wolf*, Felix Spathelf, Jan Zimmermann, Theresa Lundbeck,
Marius Peters, Kristin Kliemt, Cornelius Krellner and Michael Lang

Physikalisches Institut, J.W. Goethe-Universität Frankfurt(M),
Max-von-Laue-Str. 1, 60438 Frankfurt, Germany

* wolf@physik.uni-frankfurt.de

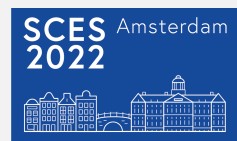 *International Conference on Strongly Correlated Electron Systems
(SCES 2022)
Amsterdam, 24-29 July 2022*

## Abstract

We present results of magnetic susceptibility and thermal expansion measurements performed on high-quality single crystals of $EuPd_2(Si_{1-x}Ge_x)_2$ for $0 \leq x \leq 0.2$ and temperatures $2\,K \leq T \leq 300\,K$. Data were taken at ambient pressure and finite He-gas pressure $p \leq 0.5$ GPa. For $x = 0$ and ambient pressure we observe a pronounced valence-change crossover centred around $T'_V \approx 160$ K with a non-magnetic ground state. This valence-change crossover is characterized by an extraordinarily strong pressure dependence of $dT'_V/dp = (80 \pm 10)$ K/GPa. We observe a shift of $T'_V$ to lower temperatures with increasing Ge-concentration, reaching $T'_V \approx 90$ K for $x = 0.1$, while still showing a non-magnetic ground state. Remarkably, on further increasing $x$ to 0.2 we find a stable $Eu^{(2+\delta)+}$ valence with long-range antiferromagnetic order below $T_N = (47.5 \pm 0.1)$ K, reflecting a close competition between two energy scales in this system. In fact, by the application of hydrostatic pressure as small as 0.1 GPa, the ground state of this system can be changed from long-range antiferromagnetic order for $p < 0.1$ GPa to an intermediate-valence state for $p \geq 0.1$ GPa.

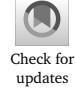

## 1  Introduction

In the search for novel collective phenomena one route of research focusses on correlated electron materials tuned to second-order critical points. As an example we mention the phenomenon of *critical elasticity* recently observed upon pressure tuning an organic Mott insulator across the second-order critical endpoint of the first-order Mott transition line [1]. On approaching the critical endpoint a pronounced softening of the lattice was revealed indicating a particular strong coupling between the critical electronic system and the lattice degrees of

freedom. The work by Gati *et al.* suggests that similar strong-coupling effects can be expected also for other critical endpoints amenable to pressure tuning. In this context, the valence transition critical endpoint represents a particularly interesting scenario as this transition involves electronic-, magnetic-, and lattice degrees of freedom. The valence transition in Eu-based intermetallics was studied intensively in the past [2]. The generic temperature-pressure (*T-p*) phase diagram derived from these studies includes $Eu^{(2+\delta)+}$ states with local magnetic moments at low pressure giving rise to long-range magnetic order at low temperatures. The energy gain associated with the formation of magnetic order among stable moments defines one relevant energy scale in the system. With increasing pressure the system crosses a first-order valence transition line $T_V(p)$, separating that high-volume $Eu^{(2+\delta)+}$ state at low pressure (and high temperatures) from a non-magnetic low-volume $Eu^{(3-\delta')+}$ state at high pressure (and low temperature) [3,4]. The energy gain associated with the valence change marks the second relevant energy scale in the system. This first-order line $T_V(p)$ terminates at a second-order critical endpoint $(T_{cr}, p_{cr})$. For some materials, a valence change can be induced just by varying the temperature. Among them is $EuPd_2Si_2$, crystallizing in the tetragonal $ThCr_2Si_2$ structure, which shows a pronounced, yet continuous valence change from $Eu^{2.8+}$ to $Eu^{2.3+}$ upon cooling through $T_V' \approx 160\,K$ [5–7]. In the valence-change crossover regime, the crossover temperature $T_V'(p)$, defined by the position where the change in the magnetic susceptibility is largest, provides a measure of the energy scale associated with the valence change. According to magnetic- [8] and thermodynamic [4] measurements on powder material, $EuPd_2Si_2$ is located on the high-pressure side of the second-order critical endpoint, i.e., in the valence-change crossover regime. This material has become of interest recently due to the availability of single crystals of pure $EuPd_2Si_2$ [3,9] and Ge-substituted $EuPd_2(Si_{1-x}Ge_x)_2$ [10] which opens up exciting possibilities for detailed investigations by using a wide range of experimental tools. The motivation of the present study has been to identify a suitable chemical modification in the series $EuPd_2(Si_{1-x}Ge_x)_2$, corresponding to $EuPd_2Si_2$ at a negative chemical pressure, so that the critical regime can be accessed via fine pressure tuning by using He-gas techniques. To this end we performed measurements of the magnetic susceptibility and the thermal expansion on selected single crystals of the series $EuPd_2(Si_{1-x}Ge_x)_2$ at ambient and finite He-gas pressure. Our study demonstrates that for $x = 0.2$ the two energy scales, determining the material's ground state, become almost degenerate. Due to their different pressure dependencies, the application of a weak pressure as small as 0.1 GPa is sufficient to alter the ground state of the material from antiferromagnetic order with $T_N$ around 47 K at low pressure to an intermediate-valence state for $p \geq 0.1$ GPa.

## 2 Experimental details

Single crystals of $EuPd_2(Si_{1-x}Ge_x)_2$ with nominal Ge-concentration $x = 0$ (#1, #2), $x = 0.1$ (#3), and $x = 0.2$ (#4, #5) were grown by using the Czochralski method. Table 1 summarizes the crystals investigated and their nominal Ge-concentration. Details of the single crystals growth and sample characterization are given in Refs. [9,10]. In what follows we refer to the crystals by their nominal Ge-concentration and specify the crystal by giving the batch number. The susceptibility was measured by using a commercial superconducting quantum interference device (SQUID) magnetometer (MPMS, Quantum Design) equipped with a CuBe pressure cell (Unipress Equipment Division, Institute of High Pressure Physics, Polish Academy of Science). The pressure cell is connected via a CuBe capillary to a room temperature He-gas compressor, serving as a gas reservoir, which enables temperature sweeps to be performed at $p \approx$ const. conditions, see Ref. [11] for details. Thermal expansion measurements were performed by using two different methods. This includes (1) a capacitive dilatometer [12,13],

Table 1: Sample number as used in this manuscript, the original batch no. together with the nominal Ge-concentrations of the single crystals investigated.

| sample no. | batch no. | nominal Ge-concentration |
|------------|-----------|--------------------------|
| **#1** | MP401_1a01 | $x = 0$ |
| **#2** | MP401_3 | $x = 0$ |
| **#3** | MP801_01a11 | $x = 0.1$ |
| **#4** | MP807_2a01 | $x = 0.2$ |
| **#5** | MP805_1a02 | $x = 0.2$ |

enabling length changes of $\Delta L \geq 5 \cdot 10^{-3}$ nm to be resolved, combined with a He-gas pressure system, see Ref. [1, 14] for details. This technique can be applied in those temperature and pressure ranges where the pressure-transmitting medium helium is in its liquid phase, i.e., above the solidification line $T_{sol}(p)$, allowing the upper capacitor plate of the dilatometer to move freely. In addition (2), a strain gauge technique in combination with the He-gas pressure system was used for measuring length changes. The strain gauge (CEFLA-1-11, Tokyo Measuring Instruments Lab.) was glued on top of the sample using a two-component adhesive (EA-2A, same manufacturer) [15]. To correct for extrinsic temperature effects and for better resolution, three strain gauges were glued on a NaCl reference and the signals of the sample and the reference were subtracted from each other *in situ* by means of a Wheatstone bridge. The resulting data were corrected for the known contribution of NaCl [16]. The strain gauge technique, which has a resolution of of $\Delta L \geq 1 \cdot 10^{-1}$ nm, allows measurements to be performed also below $T_{sol}(p)$ where helium is in its solid phase.

## 3 Results

The main panel of figure 1 exhibits the molar susceptibility $\chi_{mol}(T)$ of $EuPd_2(Si_{1-x}Ge_x)_2$ single crystals at ambient pressure ($p = 0$) for the pure compound $x = 0$ (crystal #1), and nominal Ge-concentrations $x = 0.1$ (crystal #3) and $x = 0.2$ (crystal #4) for temperatures 2 K $\leq T \leq$ 300 K. The results for $x = 0$ reproduce published data [5, 8, 17]. The susceptibility (blue circles) follows a Curie-Weiss-like increase down to about $T = 260$ K, followed by a rounded maximum and a broadened drop around 160 K, reflecting the temperature-induced valence-change crossover. Although this valence-change process spans over a considerably wide temperature range of approximately 30 K for crystal #1, we specify a crossover temperature of $T'_V \approx 160$ K by using the position of the maximum in $d(\chi \cdot T)/dT$. According to the detailed characterization work by Kliemt *et al.* [9], this crossover temperature in $EuPd_2Si_2$ may vary significantly from sample to sample depending on the actual Si concentration and internal Si-Pd ratio and disorder.

A qualitative similar behaviour, though with a significantly reduced valence-change crossover temperature of $T'_V \approx 90$ K, is revealed for crystal #3 with a nominal Ge-concentration of $x = 0.1$, cf. red circles in Fig. 1. In contrast, $\chi_{mol}(T)$ of crystal #4 with $x = 0.2$ (green circles) follows a Curie-Weiss like behaviour over an extended range from room temperature down to about 60 K, indicating a stable and temperature independent $Eu^{(2+\delta)+}$ valence under these conditions. On further cooling, the susceptibility shows a sharp kink-like anomaly at $(47.5 \pm 0.1)$ K. A similar behaviour in $\chi_{mol}(T)$ was observed also in $EuPd_2Ge_2$ [3] and

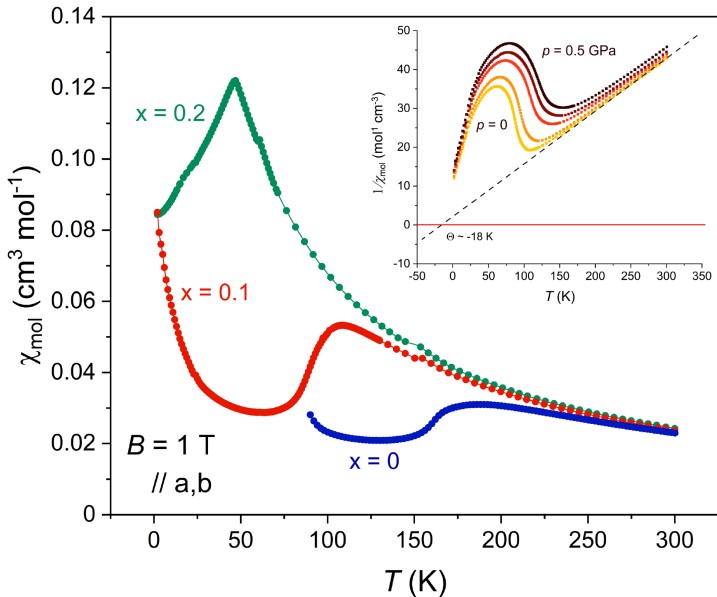

Figure 1: Molar magnetic susceptibility as a function of temperature $\chi_{mol}(T)$ at ambient pressure of $EuPd_2(Si_{1-x}Ge_x)_2$ for $x = 0$ (blue circles), $x = 0.1$ (red circles) and $x = 0.2$ (green circles). Measurements were performed with an external field of $B = 1.0$ T oriented along the tetragonal $a,b$-plane. Inset: Inverse susceptibility for $x = 0.1$ as a function of temperature in the range $2$ K $\leq T \leq 300$ K and varying hydrostatic He-gas pressure values of 0, 0.01, 0.3, 0.4 and 0.5 GPa.

$Eu(Pd_{1-x}Au_x)_2Si_2$ for $x = 0.2$ and 0.25 [8] where it was assigned to an antiferromagnetic transition. In fact, a full characterization of the magnetic and thermodynamic properties of $EuPd_2(Si_{1-x}Ge_x)_2$ for $x = 0.2$ by means of magnetization and specific heat performed in Ref. [10], provides clear evidence for a phase transition at $T_N = (47.5 \pm 0.1)$ K into long-range antiferromagnetic order. As we will demonstrate below, this assignment is consistent with the response obtained under hydrostatic pressure.

We now turn to the results obtained under finite hydrostatic pressure. In the inset of figure 1 we show the inverse susceptibility of a single crystal with $x = 0.1$ (#3) as a function of temperature for hydrostatic He-gas pressures ranging from $p = 0$ (yellow squares) up to 0.5 GPa (dark brown squares). This representation allows to extract a Curie-Weiss temperature of $\Theta_{CW} = -(18 \pm 0.5)$ K from a fit to the experimental data at $p = 0$ in the temperature range $150$ K $\leq T \leq 300$ K, where $1/\chi_{mol}(T)$ varies linearly with temperature. This linear variation in $1/\chi_{mol}(T)$ is preserved also for the data at higher pressure, albeit for a somewhat restricted temperature range, and a small increase in $|\Theta|$ with increasing pressure. As observed for the $x = 0$ compound (crystals #1 and #2), there is no hysteresis revealed upon cooling and warming, indicating that the $x = 0.1$ compound is still in the valence-change crossover regime, i.e., on the high-pressure side of the second-order critical endpoint. The valence-change crossover regime, clearly visible in the $1/\chi$ vs. $T$ representation as bended curves, grows with increasing pressure. By identifying the crossover temperature with the maximum in $d(\chi \cdot T)/dT$ we obtain an extraordinarily large pressure dependence of $dT_V'/dp = (80 \pm 10)$ K/GPa.

Figure 2 (main panel) displays the relative length changes $\Delta L(T)/L_0$, with $\Delta L = L(T) - L_0$ and $L_0$ the sample length at a reference temperature, typically the temperature at the beginning of the experiment, measured along an axis within the tetragonal plane of single crystalline

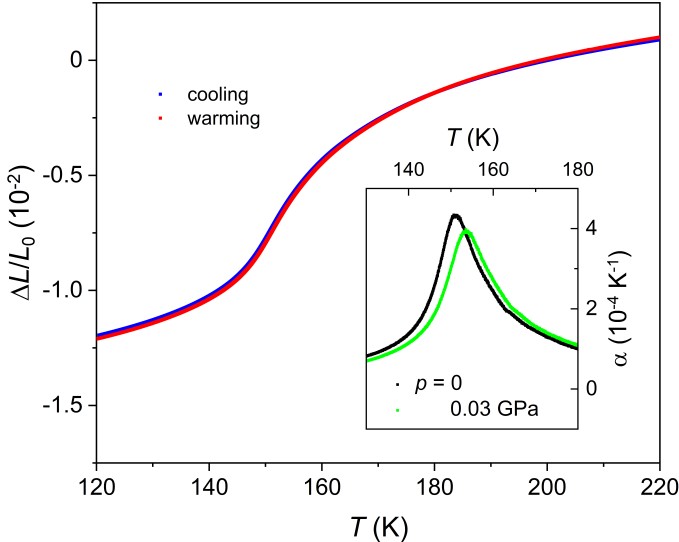

Figure 2: Relative length changes of single crystalline $EuPd_2Si_2$ (crystal #2) in the tetragonal plane around the valence-change crossover for decreasing (blue line) and increasing (red line) temperature. Inset: Coefficient of thermal expansion $\alpha(T)$ in the temperature range 150 K $\leq T \leq$ 180 K at ambient pressure (black line) and at $p = 0.03$ GPa (light green line).

$EuPd_2Si_2$ (crystal #2) for temperatures 120 K $\leq T \leq$ 220 K. The data reveal a pronounced anomaly, i.e., a broadened shrinkage of the in-plane lattice parameter $\Delta a/a$ of order -0.7 $\cdot 10^{-2}$ upon cooling through the valence-crossover regime 130 K $\leq T \leq$ 180 K. The data taken upon cooling (blue) and warming (red) lack any indication for hysteretic behaviour within the experimental resolution. The broadened anomaly and the absence of any thermal hysteresis indicate that single crystalline $EuPd_2Si_2$ is located on the high-pressure side of the second-order critical endpoint in the valence-change crossover regime, consistent with the susceptibility data in Fig. 1 and earlier results taken on powder material [4]. The inset of figure 2 exhibits the coefficient of thermal expansion $\alpha = 1/L \cdot (\partial L/\partial T)_p$ as a function of temperature for $EuPd_2Si_2$ (crystal #2) at $p = 0$ (black symbols) and $p = 0.03$ GPa (green symbols). By assigning the maximum in $\alpha$ to the valence-change crossover temperature, we find $T'_V = (151 \pm 0.1)$ K for $p = 0$. For the data at $p = 0.03$ GPa, we observe a significant shift in the position of this maximum to higher temperature of $T'_V = (153.5 \pm 0.1)$ K which is accompanied by a mild reduction in the size of the maximum. From these measurements at $p = 0$ and 0.03 GPa, we estimate a pressure dependence of the valence-change crossover temperature of $dT'_V/dp = (80 \pm 10)$ K/GPa. This value is identical to the one derived above from the susceptibility data under pressure for $x = 0.1$ (crystal #3).

Figure 3 shows the magnetic susceptibility of $EuPd_2(Si_{1-x}Ge_x)_2$ with $x = 0.2$ (crystal #4) (Fig. 3a) together with the coefficient of thermal expansion for $x = 0.2$ (crystal #5) (Fig. 3b) as a function of temperature for $p = 0$ and various pressures up to 0.4 GPa. The inset of figure Fig. 3b exhibits a $p$-$T$ phase diagram displaying $T_N(p)$ (dark-red open squares) and $T_{max}(p)$ (blue-open squares). The antiferromagnetically ordered state and the valence-change crossover regime are indicated by the shaded orange area and the shaded blue area, respectively. With respect to the antiferromagnetic transition temperature $T_N$ at $p = 0$, the data reveal a small difference of about 2 K between crystals #4 and #5. We attribute this difference to small variations of the crystals' Ge-concentrations, for details see Refs. [9, 10]. As discussed above, the sharp kink-like anomaly in the magnetic susceptibility can be assigned

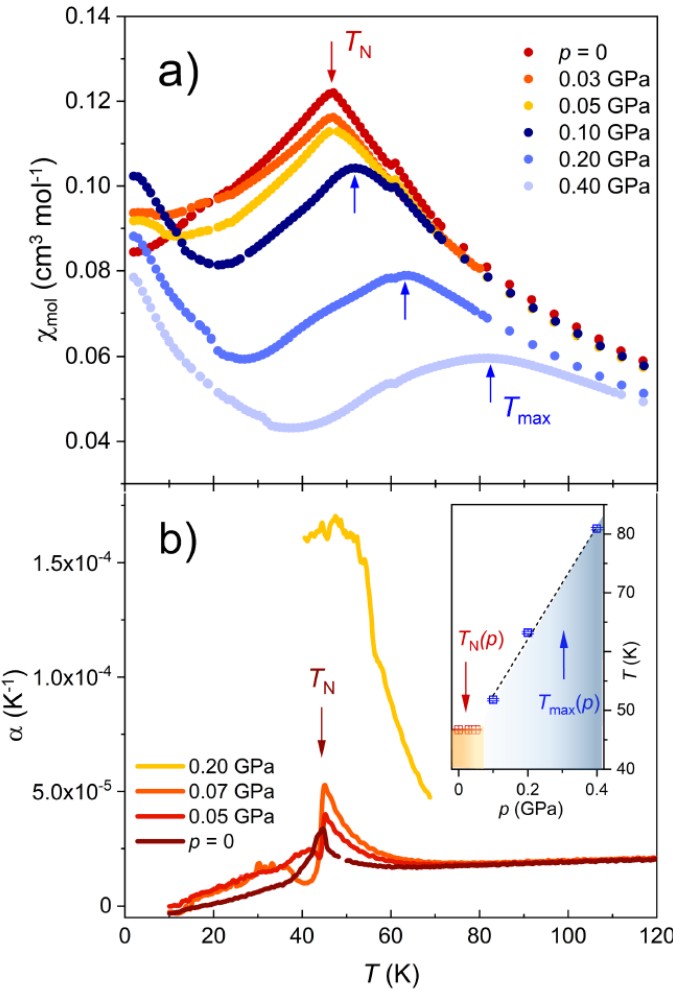

Figure 3: : **a)** $\chi_{mol}(T)$ of $EuPd_2(Si_{1-x}Ge_x)_2$ for $x = 0.2$ (crystal #4) at ambient pressure $p = 0$ (dark red full circles), $p = 0.03$ GPa (orange full circles), and $p = 0.05$ GPa (yellow full circles). The sharp kink in $\chi_{mol}(T)$ at $p = 0$, marked by the dark red downward arrow, is assigned to an antiferromagnetic transition at $T_N = (47.5\pm0.1)K$. The various data sets coloured in different blue shades represent $\chi_{mol}(T)$ taken at $p \geq 0.1$ GPa, with 0.1 GPa (dark blue), 0.2 GPa (azure blue) and $0.4GPa$ (light blue). The blue upwards arrows, labelled by $T_{max}$, mark the position of the broad maximum in $\chi_{mol}(T)$. We assign $T_{max}$ to the onset of the valence-change crossover regime. **b)** Coefficient of thermal expansion measured along the tetragonal plane of $EuPd_2(Si_{1-x}Ge_x)_2$ for $x = 0.2$ (crystal #5) as a function of temperature. The data are taken in the range $10$ K $\leq T \leq 100$ K and hydrostatic He-gas pressures from $p = 0$ (brown symbols), 0.05 GPa (red symbols), 0.07 GPa (orange symbols) and 0.2 GPa (yellow symbols). **inset)** $p$-$T$ phase diagram displaying $T_N(p)$ (dark-red open squares) and $T_{max}(p)$ (blue- open squares). See text for details.

to the phase transition into long-range antiferromagnetic order below $T_N = (47.5 \pm 0.1)$ K. Upon applying He-gas pressure of $p = 0.03$ GPa (orange full circles) and $p = 0.05$ GPa (yellow full circles), the character of the transition in $\chi_{mol}(T)$ is retained, except a reduction of the susceptibility at higher temperatures and around $T_N$ and a small slight rounding of the kink anomaly at $T_N$. In addition, we find indications for a weak upturn of $\chi_{mol}(T)$ at the low-temperature end for $p = 0.03$ GPa which becomes larger at $p = 0.05$ GPa. The data reveal

only a tiny shift of the peak position to higher temperatures with increasing pressure at a rate $dT_N/dp = (1 \pm 0.2)$ K/GPa. This value is typical for many antiferromagnetic EuT$_2$X$_2$ compounds with a transition metal T and X = Si or Ge. For example, it is similar to the pressure dependence of $T_N$ reported for EuRh$_2$Si$_2$ [3]. The situation changes drastically upon further increasing the pressure to $p = 0.1$ GPa, 0.2 GPa and 0.4 GPa. In this pressure range the sharp kink-like anomaly in $\chi_{mol}(T)$ is replaced by a broadened maximum, labelled $T_{max}$ in Fig. 3a), followed by a reduction of $\chi_{mol}(T)$ and a pronounced upturn at lower temperatures. This behaviour is reminiscent to $\chi_{mol}(T)$ revealed for the compounds with $x = 0$ and 0.1, reflecting a temperature-induced valence-change crossover for $p \geq 0.1$ GPa. In fact, this assignment is corroborated by looking at the pressure dependence of $T_{max}$. We find $T_{max} = (52.1 \pm 0.2)K$ for $p = 0.1GPa$ which increases to $T_{max} = (63.3 \pm 0.2)$ at K (0.2 GPa) and $T_{max} = (82.4 \pm 0.2)K$ (0.4 GPa), which corresponds to a pressure dependence of $dT_{max}/dp = (145 \pm 45)$ K/GPa. This value, which is distinctly different from d $T_N/d\ p = (1.0 \pm 0.2)$ K/GPa revealed for low pressures $p \leq 0.05$ GPa, is of the same order of magnitude as $dT'_V/dp$ observed for the $x = 0$ and 0.1 compounds. We therefore conclude that hydrostatic He-gas pressure as small as $0.1GPa$ is sufficient to change the ground state of the $x = 0.2$ compound from long-range antiferromagnetic order at low pressures to a state governed by a valence-change crossover for $p \geq 0.1$ GPa. This interpretation for $p \geq 0.1GPa$ is further corroborated by preliminary results of the coefficient of thermal expansion obtained for $x = 0.2$ (crystal #5) by using strain gauges, shown in Fig. 3b). For the data at $p = 0$, we find a gradual decrease in $\alpha(T)$ on cooling the crystal from 100 K down to about 60 K. Upon further cooling, the data reveal an increase of $\alpha(T)$ followed by a peak anomaly around 45 K. We assign this feature to the phase transition into long-range antiferromagnetic order. For measurements under finite He-gas pressure of $p = 0.05$ GPa (red full line) and 0.07 GPa (full orange line), we observe a sharp jump-like anomaly, the position of which remains practically unchanged in this pressure range. The insensitivity to weak hydrostatic pressure is consistent with this feature reflecting the transition into long-range antiferromagnetic order. However, the data at finite pressure reveal an additional contribution to the thermal expansion, setting in distinctly above $T_N$, which grows with increasing pressure. In order to understand the origin of this contribution, it is enlightening to look at the data obtained on further increasing the pressure to $p = 0.2$ GPa, where according to the susceptibility data (Fig. 3a) the system has entered the valence-change crossover regime. In fact, at $p = 0.2$ GPa the $\alpha(T)$ data show an extraordinarily large positive contribution with a maximum around 50 K, the size of which now reaching values of about $1.7 \cdot 10^{-4}$ K$^{-1}$. This size is of the same order of magnitude as observed for the valence-change crossover for the $x = 0$ compound, cf. inset to Fig. 2. These observations at $p = 0.2$ GPa suggest that the positive contribution to $\alpha(T)$, visible already at low pressures upon approaching $T_N$ from above, signals the growing tendency of the system towards a valence change. We like to point out that in attempts to reproduce the pronounced anomaly around 50 K, we have observed that thermal cycling leads to different curves, which prevented us from extending this first set of measurements for covering a wider temperature range. We attribute these effects to the extraordinarily large thermal expansivity accompanying the valence change. This has to be taken into account in future thermal expansion experiments on this system.

## 4  Conclusion

Using magnetic susceptibility and thermal expansion measurements under He-Gas pressure on high-quality single crystals of EuPd$_2$(Si$_{1-x}$Ge$_x$)$_2$ for $0 \leq x \leq 0.2$ we were able to investigate the interplay between valence fluctuations and long-range antiferromagnetic order. At ambient pressure, the compound with $x = 0$ is in the valence-change crossover regime with

a characteristic temperature of $T'_V \approx 160$ K. Upon increasing the Ge concentration to $x = 0.1$ $T'_V$ becomes significantly reduced to about 90 K, indicating a weakening of the energy scale associated with the valence change with increasing x. In fact, on further increasing x to 0.2, the tendency towards valence fluctuations is further suppressed. Instead the magnetic data indicate a stable and temperature-independent $Eu^{(2+\delta)+}$ valence accompanied by long-range antiferromagnetic order below about $T_N = 47$ K. These findings suggest that for $x = 0.2$ the two energy scales associated with these two competing ground states, determining the material's ground state, are almost identical with a slight advantage for local moment magnetic order. This becomes particularly clear by experiments under finite hydrostatic pressure. Due to the distinctly different response these energy scales show to the application of hydrostatic pressure, as reflected by $dT'_V/dp = (80 \pm 10)$ K/GPa *vs* $dT_N/dp = (1.0 \pm 0.2)$ K/GPa, the ground state for $x = 0.2$ can be changed from long-range antiferromagnetic order for $p < 0.1$ GPa to an intermediate valence state for $p \geq 0.1$ GPa.

## Acknowledgements

Funded by the Deutsche Forschungsgemeinschaft (DFG, German Research Foundation) - TRR 288 - 422213477 (projects A01 and A03).

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
