# Peer review of "From magnetic order to valence-change crossover in EuPd$_2$(Si$_{1-x}$Ge$_x$)$_2$ using He-gas pressure"

_SciPost Physics Proceedings, doi:SciPost Phys. Proc. 11, 022 (2023)_

## Round 1 · Referee Report · Anonymous (Referee 1) · 2022-10-4

Strengths

1) the paper reports on new experimental data on the valence crossover in EuPd2(Si1-xGex)2 systems by measures of the magnetic susceptibility and thermal expansion at ambient and applied pressures. 2) The paper is clearly written and the results are pertinent.

Weaknesses

1) I recommend to put the legend in figs 3a) and fig 3). It is annoying to be forced to read the figure to see the legend. 2) The paper would profit from a plot of the doping, pressure, temperature phase diagram. I strongly recommend this for publication.

Report

As stated above the paper is clearly written and deserves publication in the proceedings of the SCES conference. However, I strongly recommend to show a phase diagram indicating the thermal expansion and susceptibility data under pressure.

Requested changes

1) legend in fig 3 2) additional figure (or as insert) of phase diagram indicating the phase lines AF, valence transition as function of pressure, concentration.

  • validity: good
  • significance: good
  • originality: good
  • clarity: good
  • formatting: excellent
  • grammar: perfect

Author:  Bernd Wolf  on 2022-10-21  [id 2942]

(in reply to Report 1 on 2022-10-04)

---

## Round 1 · Referee Report · Anonymous · 2022-10-4

„From magnetic order to valence-change crossover in EuPd2(Si1-xGex)2 using He-gas pressure”
by B Wolf, et al.

report:
The paper reports on new experimental data on the valence crossover in EuPd2(Si1-xGex)2 systems by measures of the magnetic susceptibility and thermal expansion at ambient and applied pressures. The paper is clearly written and the results are pertinent and deserves publication in the proceedings of the SCES conference. However, I strongly recommend to show a phase diagram indicating the thermal expansion and susceptibility data under pressure. Furthermore, I recommend to put the legend in fig. 3a) and fig. 3b).

We thank the Referee for reading the manuscript and for his/her comments and recommendations. As suggested by the referee, we added legends in fig. 3a and fig. 3b. In addition, we provide an inset in fig. 3b which displays a p-T phase diagram of EuPd2(Si1-xGex)2 and we modified the figure captions accordingly. Two sentences have been added at the beginning of the last paragraph on page 5 and at the end of page 5 where we refer to the new inset of figure 3b.
To facilitate the work by the Referee and the Editor, we provide, in addition to the revised version, a PDF version of the revised manuscript, where we indicate all the changes made. We believe that by addressing the points raised by the Referee the revised paper is now ready for publication.

Attachment:

letter_editor-20102022-BW-ML.pdf

---

## Round 2 · Author Response

---------------------------------------------------------------------- Anonymous Report 1 on 2022-10-4 (Invited Report) for scipost_202207_00023v1 „From magnetic order to valence-change crossover in EuPd2(Si1-xGex)2 using He-gas pressure” by B Wolf, et al. ----------------------------------------------------------------------

System Message: ERROR/3 (<string>, line 5)

Unexpected indentation.

report:

System Message: WARNING/2 (<string>, line 6)

Block quote ends without a blank line; unexpected unindent.

The paper reports on new experimental data on the valence crossover in EuPd2(Si1-xGex)2 systems by measures of the magnetic susceptibility and thermal expansion at ambient and applied pressures. The paper is clearly written and the results are pertinent and deserves publication in the proceedings of the SCES conference. However, I strongly recommend to show a phase diagram indicating the thermal expansion and susceptibility data under pressure. Furthermore, I recommend to put the legend in fig. 3a) and fig. 3b). comments: We thank the Referee for reading the manuscript and for his/her comments and recommendations. As suggested by the referee, we added legends in fig. 3a and fig. 3b. In addition, we provide an inset in fig. 3b which displays a p-T phase diagram of EuPd2(Si1-xGex)2 and we modified the figure captions accordingly. Two sentences have been added at the beginning of the last paragraph on page 5 and at the end of page 5 where we refer to the new inset of fig ure 3b.

---

## Round 2 · List of Changes

1.) added legends in fig. 3a and fig. 3b.
2.) provide an inset in fig. 3b.
3.) we modify the figure caption of fig.3
4.) added two sentences at the beginning of the last paragraph on page 5 and at the end of page 5 where we refer to the new inset of fig ure 3b.

---

## Editorial Decision

published